# Influence of Climate and Local Habitat Characteristics on Carabid Beetle Abundance and Diversity in Northern Chinese Steppes

**DOI:** 10.3390/insects11010019

**Published:** 2019-12-24

**Authors:** Noelline Tsafack, Yingzhong Xie, Xinpu Wang, Simone Fattorini

**Affiliations:** 1School of Agriculture, Ningxia University, 489 Helanshan West Road, Yinchuan 750021, China; xieyz@nxu.edu.cn; 2Department of Life, Health, and Environmental Sciences, University of L’Aquila, 67100 L’Aquila, Italy; simone.fattorini@univaq.it

**Keywords:** central Asia, climate factors, abundance, diversity, dominance, eigenvector spatial filtering, environmental severity

## Abstract

Carabids are an important insect group in grassland ecosystems and are involved in numerous ecosystem services. Steppes are the most widespread ecosystems in China, but they are under increasing degradation. Despite their importance, little is known about the relationships between environmental variables and carabid community structure in Chinese steppes. We studied the effects of fine-scale factors (soil and vegetation) and coarse-scale factors (climate) on carabid community parameters (abundance, richness, diversity, dominance, and evenness) in three types of steppes (desert, typical, and meadow steppes) in northern China. Carabid communities responded to environmental factors in different ways according to the type of steppe. Climate factors were the most important drivers of community structure, whereas the effects of soil and vegetation were less important. Desert steppe showed the lowest carabid abundance, richness, diversity, and evenness, and the highest dominance. This community is relatively simple and strongly dominated by a few species adapted to the severe conditions of this environment. Typical and meadow steppes showed carabid communities with a more complex structure. As expected on the basis of environmental severity, the most severe ecosystem (i.e., the desert) was only influenced by climatic factors, whereas a certain influence of biotic factors emerged in the other ecosystems.

## 1. Introduction

Steppes are the most widespread ecosystems in China. They are classified into four major types, (called “meadow steppe”, “typical steppe”, “desert steppe”, and “alpine steppes”, respectively [1]). Covering more than 40% of China’s land surface, they have immense ecological and socioeconomic importance [2,3]. Unfortunately, Chinese steppes are degrading rapidly due to climate change and human activities that are changing productive grasslands into barren land and desert [4,5]. These degradation processes will have tremendous impacts on steppe biodiversity, which calls for urgent research on the ecology of groups that may serve as model organisms.

Carabid beetles (Coleoptera: Carabidae) constitute a prominent group of ground-dwelling insects in grassland ecosystems and are involved in numerous ecosystem services and functions, playing direct or indirect important roles in soil formation and structure, energy flow, and nutrient cycling [6,7,8]. They also contribute to pest regulation [9,10], and serve as food for a wide range of animals [11]. Moreover, carabids are known to be particularly sensitive to habitat alteration, and so are frequently used in studies of anthropogenic changes to the landscape [12,13,14,15]. However, despite the importance of carabids in steppe ecology [5], little is known about the relationships between environmental variables and carabid community structure in Chinese steppes.

In a previous study, we investigated the effects of local habitat characteristics on carabid species functional diversity in three types of steppes (desert, typical, and meadow [3]) in northern China [16]. We found that, in general, temperature, precipitation, and humidity were positively related to functional diversity at the regional scale, but important differences emerged between the different types of steppes. For example, temperature was important in promoting functional diversity in the typical and in the meadow steppes, but not in the desert steppe (where temperature can reach very high values), whereas humidity was important in the desert and the typical steppe but not in the meadow steppe. We also found that soil temperature had negative effects on carabid functional diversity at the regional scale.

In the present study, we focused on the effects of fine-scale factors (such as soil properties and vegetation characteristics) and coarse-scale factors (i.e., climate) on various aspects of carabid community organization (diversity, dominance, and evenness) in the same three types of steppes (desert, typical, and meadow).

Specifically, we predicted the following influences of fine- and coarse-scale factors on carabid community organization:

(1) Climate. Humidity and precipitation should affect carabid community structure, because humidity is one of the most important abiotic drivers of carabid communities [12,17] and water is scarce in steppes. Temperature should be positively related to activity density and diversity in the typical and meadow steppes, because higher temperatures favor the activity of many carabid species [18,19], at least until they become too high. In fact, in the desert steppe, where temperatures can be extremely high and humidity is low, only few thermophilic species can survive [20], and their activity should be inhibited by too high temperatures.

(2) Soil compaction. Bulk density (a proxy of compaction) should negatively impact carabids because soil compaction could make egg-laying and burrowing difficult [21].

(3) Soil temperature. Higher soil temperatures should favor carabids in the meadow and typical steppes, but increasing temperatures should have a negative influence in the desert steppe, where very high temperature values can be reached.

(4) Soil litter and moisture. The amount of litter and moisture should favor carabids, because they improve soil fertility, food availability, and habitat heterogeneity [22,23].

(5) Vegetation. Carabids should be favored (and hence attain higher abundance) in habitats with higher values plant biomass, density, height, and diversity, because more abundant, diversified, and complex (multilayer) vegetation should provide carabids with more microhabitats (including more food and shelter sites) [24,25,26,27,28].

In general, we expect that the carabid community of the desert steppe should be composed of relatively few species, strongly adapted to the harsh conditions of this environment. We hypothesize, therefore, that the desert should have communities with strong dominance, and lower diversity and evenness, compared to those of the typical and meadow steppes. By using indices of abundance, richness, diversity, dominance, and evenness, we investigated how these different aspects of community structure respond to the same environmental characteristics. In general, since no single index can adequately describe all aspects of community structure, it is important to use a combination of them that reflects diversity, dominance, and evenness [29,30,31].

There is a long debate about the relative importance of biotic and abiotic factors in determining community structure. In general, it is supposed that abiotic parameters of the environment become less and less important in determining community structure while the role of biotic characteristics increases, as environmental severity decreases [32,33,34,35]. Thus, we can expect that abiotic factors (such as climate, soil compaction, and soil temperature) should be more important in the most severe ecosystem (i.e., the desert steppe), whereas biotic factors (such as vegetation characteristics) should be more important in the less severe ecosystems (i.e., the typical and meadow steppe).

## 2. Materials and Methods 

### 2.1. Study Areas and Data Collection

We selected three sampling areas in the Ningxia region (northern China) representative of different grassland ecosystems [3]:

(1) A desert steppe, characterized by a cold, semi-arid continental monsoonal climate, with a mean annual temperature of 5.7 °C (mean monthly temperature ranging from −8 °C in January to 22 °C in July) and mean annual precipitation of 200 mm [3]. Vegetation is mainly represented by drought-tolerant species, such as *Agropyron mongolicum*, *Artemisia desertorum*, *A. blepharolepi*, and *Stipa* spp.

(2) A typical steppe characterized by a continental monsoon climate with a mean annual temperature of 5.7 °C (mean monthly temperature ranging from −22 °C in January to 28 °C in August) and mean annual precipitation of 350 mm [3]. In this mountain area, to reflect possible within-ecosystem variability, we selected three sectors according to the main vegetation types: (i) A first sector, at the top of the mountain, occupied by natural patches of grass vegetation (including *Stipa bungeana*, *S. grandis*, *Artemisia frigida*, *Thymus mongolicus*, and *Heteropappus altaicus* as dominant species); (ii) a second sector, also at the top of the mountain, but with vegetation crossed by patches of cut grasses that served as fire belts; and (iii) a third sector, at the bottom of the mountain, with a mosaic of crops and natural vegetation (including *S. bungeana*, *A. frigida*, *T. mongolicus*, and *Potentilla acaulis* as dominant species).

(3) A meadow steppe, characterized by a semi-humid climate, with a mean annual temperature of 7 °C (mean monthly temperature ranging from −7 °C in January to 20 °C in July) and mean annual precipitation of 450 mm [3]. To reflect possible within-ecosystem variability, we selected two sampling sectors, at the south-west side of the mountain peak (dominated by several species of the genus of *Festuca*, principally the alpine fescue *F. brachyphylla*) and at the bottom of the mountain peak (dominated by *S. bungeana*, *A. frigida*, and *Achnatherum splendens*), respectively.

We selected 15 sampling sites in the desert steppe (distributed in an area of about 1 km^2^), 45 sites (15 sites per sector) in the typical steppe (distributed in an area of about 3 km^2^), and 30 sites (15 sites per sector) in the meadow steppe (distributed in an area of about 2 km^2^), for a total of 90 sampling sites. The distance between the sampling areas in three ecosystems (desert, typical, and meadow steppes) were as follows: The desert was about 250 km from the meadow and the typical steppe; the typical and the meadow steppe were about 70 km apart. Thus, there is no risk that some sampling points would trap species of adjacent habitats.

At each sampling site (placed at random and separated by at least 150 m from each other to minimize spatial autocorrelation), five pitfall traps (separated by at least five meters from each other) were used. Pitfall traps consisted of plastic cups (diameter: 7.15 cm, depth: 9 cm) covered by a transparent plastic lid and sunk in the ground with the cup-lip level with the soil surface. Each was filled with 60 mL of a mixture of tap water and vinegar (8%) and sugar (4%) as attractant and 70% alcohol (4%) as preservative.

Sampling was conducted in 2017, from May to September, which corresponds to the main period of carabid activity in the study ecosystems [36]. Lövei and Magura (2011) [37] examined the influence of two components of the trapping effort on carabid collection: The number of traps and the time for which they are active. They found that the two components are not equivalent and that it is better to increase the number of traps and shorten the time of sampling than using fewer traps for longer periods. Following Lövei and Magura’s (2011) [37] suggestion, we adopted a sampling design using many traps for a short period. Thus, during the sampling period, pitfall traps were set once a month mid-month, and left in the field for 72 h prior to emptying and removal of the traps. Traps could not operate for more than 72 h, because in this time span, they became completely full of beetles. Thus, we collected 25 samples (5 pitfall traps × 5 sampling dates) for each site (90 sampling sites), for a total of 2250 samples, over a period of five months, which allowed us to have a good representation of carabid richness and abundance across habitats. This sampling strategy was also used to assure that local beetle populations were not over-sampled. The captured carabids were counted and identified to species based on keys and museum specimens with the aid of a taxonomist expert in carabid beetles (Prof. H. Liang, see Acknowledgments). In all analyses, we pooled the data from the same site, because soil and vegetation characteristics were observed at the sampling site level. No trap was lost during the sampling. In all analyses dealing with species abundance, we used species’ activity density, calculated as the number of individuals from each species divided by the number of traps active in each site.

At each sampling site (25 samples × 90 sampling sites), within a 1 m^2^ quadrat frame, we measured: Plant dry biomass (PB, g/m²), cover (PC, % of soil covered by plants), density (PD, number of plants per m²), height (PH, average, cm), diversity (expressed as species richness, PSD), litter dry mass (SL, g/m²), soil moisture (SM, %), bulk density (SBD, g/cm^3^), and soil temperature (ST, 10 cm depth). We used only one quadrat per site because the habitat around each site is very homogeneous. Also, since some sites were relatively close, using more quadrats would increase the risk of autocorrelation in vegetation data. Monthly mean values of humidity (Hum), precipitation (Prec), and temperature (Temp) were recorded from meteorological stations. Further details about the study area and data collection can be found in a companion paper [16].

### 2.2. Data Analysis

We used various indices of community structure (namely richness, diversity, dominance, and evenness) to compare communities of different grassland types. However, for each of these aspects of community structure, different indices have been proposed, and there is no consensus about which index best describes a certain aspect of community structure. Thus, we followed recommendations provided by recent comparative analyses [29,30,31] and then focused on those that had the best explanatory value for our data set. The most obvious descriptor of community structure is the number of species. However, the observed values of species numbers are influenced by sampling effort, thus they cannot represent the true species richness of a community. Nonparametric estimators use the species–abundance and/or occurrence relationships throughout the samples to estimate the total number of species, using a previously formulated nonparametric model. As a measure of estimated (true) species richness, we used the bias-corrected form of the Chao-1 nonparametric estimator, which is an asymptotic estimator based on the number of singletons and doubletons. Comparative analyses of the behavior of nonparametric estimators showed that *Chao-1* performed best [38,39].

In addition to the Chao-1 estimator, we used the Margalef richness index (*Mg*). This index seems to have a good discriminant ability [40] but is strongly influenced by the sampling size and effort [31]. Since we calculated values of indices for each site independently and in each site, we had the same sampling effort, and it is likely that our values are not biased.

To take into account relative abundances, we used three indices of diversity that use the number of collected individuals for each species: The Shannon–Wiener, the exponential Shannon–Wiener, and Brillouin indices. The Shannon–Wiener index (*H*’) ranges from 0 (one species dominates the community completely) to high values for communities with many species, each with a similar number of individuals. *H*’ has its foundations in information theory and represents the uncertainty about the identity of an unknown individual [41]. *H*’ is sensible to sample size and its dependence on richness makes it difficult to compare communities with large richness differences. On the other hand, by combining richness and evenness, this index is considered particularly effective in encapsulating different aspects of diversity into a single value [42]. We also used the so-called exponential Shannon–Wiener index, e*^H’^*, which is equivalent to the number of equally common species required to produce the value of *H’* given by the sample [29,31]. It corresponds to the so-called Hill’s number *N_1_* [42]. Magurran [31] suggests the use of the Brillouin (*HB*) index in situations where randomness of the sampling is not guaranteed or if the community is completely censused. Thus, the Brillouin index measures the diversity of a collection, as opposed to the Shannon–Wiener index, which measures a sample. When the species differ in their capture rate, which may be the case of carabids with possibly different dispersal ability or that might be differently attracted by the pitfall traps [43], randomness of the sampling is not guaranteed, and this information measure might be better than the Shannon–Wiener index. 

As measures of dominance, we used the Simpson (*D*) and the Berger–Parker (*d*) indices. The Simpson index represents the probability that two randomly chosen individuals belong to the same species [41]. *D* varies from 0 (all species are equally present) to 1 (one species dominates the sample completely). The Berger–Parker index is simply the relative abundance of the most abundant species. Simpson’s index provides a good estimate of diversity at relatively small sample sizes and, in essence, it captures the variance of the species abundance distribution, whereas the Berger–Parker index does not have restrictions on the sample size [31] and is considered to be one of the best measures of dominance [44]. Both *d* and *D* increase when diversity in the community decreases. The multiplicative inverses of *D* and *d* correspond, respectively, to the Hill’s numbers, *N*_2_ (*N*_2_ = 1/*D*) and *N_∞_* (*N_∞_* = 1/*d*), which are measures of diversity [42]. *N*_1_ places more emphasis on rare than common and abundant species; *N*_2_ places more emphasis on rare and less on abundant than *N*_1_; and *N_∞_* expresses the influence of the most dominant species [42]. Thus, for comparison purposes, we used also these indices.

Finally, as a measure of evenness, we used Pielou’s index of equitability (*J*), which is the ratio between *H’* and the logarithm of species richness. This index measures the evenness with which individuals are divided among the species present, and is constrained between 0 (one species dominates the community completely) and 1 (all species are equally abundant) [29].

Although there are conceptual links between dominance, diversity, and evenness (dominance is inversely correlated with both evenness and diversity, and evenness and diversity are positively correlated), they are not interchangeable and, when considering complex interactions, the choice of index can profoundly alter the interpretation of results [41]. Simultaneously considering multiple indices can provide greater insight into the interactions in a system [41]. Some redundancy is, however, expected and, in some cases, obvious (such as in using *D* and *N*_2_, or d and *N_∞_*).

Differences in carabid community parameters (richness, abundance, diversity, dominance, and evenness indices) among the three grassland types were tested using a nested analysis of variance (nested ANOVA, with the type of grasslands as the fixed effect and sub-types of grassland as the random effect), followed by post hoc Tukey tests using the “multcomp” package in R [45]. The effects of vegetation, soil, and climate characteristics on carabid community parameters were investigated using random-effect eigenvector spatial filtering (RE-ESF) [46] with the “spmoran” package in R [46] to take into account spatial dependence. Preliminary to RE-ESF analyses, variance inflation factors (VIF) were calculated using the “usdm” package [47] to detect possible collinearity between explanatory variables. A high VIF (>10) indicates that the predictor is strongly dependent on others and does not carry independent information. Since we detected a VIF = 10.65 for ST in the desert steppe (Appendix A), this variable was no longer considered in the analyses for this ecosystem. Further details about this type of analysis can be found in a companion paper [16].

Based on the determination coefficients (r^2^), the Chao 1 index was better suited than the Margalef index as a richness index (Appendix A); the Brillouin index was better than the Shannon–Wiener index (Appendix A) and the exponential Shannon–Wiener index (Appendix A) as a diversity index. The Simpson index was better than the Berger–Parker index as a dominance index (Appendix A). Moreover, the Brillouin index also showed higher r^2^-values than the multiplicative inverses of Simpson (Appendix A) and Berger–Parker (Appendix A) indices. We therefore concentrated on these indices.

## 3. Results

### 3.1. Comparisons of Community Organization Between Ecosystems

A total of 6873 individuals belonging to 25 species of carabids were collected [48]. Seven species were common to the three types of steppe (*Carabus glyptoterus*, *Carabus vladimirskyi*, *Calosoma chinense*, *Pseudotaphoxenus rugupennis*, *Zabrus potanini*, *Amara dux*, and *Amara harpaloides*) and eight were exclusively found in one type of steppe (*Cymindis binotata*, *Corsyra fusula*, *Amara helva*, and *Harpalus lumbaris* found only in the desert steppe; *Dolichus halensis* found only in the typical steppe; and *Carabus modestulus*, *Carabus gigoloides*, and *Carabus crassesculptus* found only in meadow steppe). *Carabus vladimirskyi* was the most abundant species in the typical steppe and in the meadow steppe, accounting for 35% and 30% of the collected carabids, respectively. By contrast, *C. glyptoterus* was the most abundant species in the desert steppe, accounting for 75% of the total carabid abundance in this grassland type. The desert steppe was the grassland type with the lowest activity density, whereas no significant difference was found between the meadow and the typical steppes (Appendix A). The desert steppe was also the grassland with the highest dominance and the lowest richness, diversity, and evenness values, whereas the meadow and the typical steppes did not differ significantly for any of the indices of dominance, richness, diversity, and evenness (Appendix A).

### 3.2. Influence of Environmental Parameters

#### 3.2.1. Species Activity Density

Carabid activity density (Table 1) was positively related to the monthly mean temperature in the typical and meadow steppes. The effect of monthly mean precipitation was positive in the typical steppe but negative in the desert steppe. None of the soil or vegetation factors had a significant effect on activity density, regardless of the type of grassland, except plant biomass, which showed a positive effect in the meadow steppe.

#### 3.2.2. Richness

Richness (Table 2) was positively influenced by temperature, in the typical steppe and in the meadow steppe, whereas no significant effect was found for the desert steppe. Humidity had a positive influence on the estimated richness in the desert steppe and in the typical steppe but not in the meadow steppe. Precipitation influenced the estimated richness in the desert steppe, and the effect was negative. Soil bulk density and soil temperature had negative effects in the typical steppe. 

#### 3.2.3. Diversity

Diversity (Brillouin index, Table 3) was positively influenced by temperature in the typical and meadow steppes. Humidity had a positive effect in the desert and typical steppes but not in the meadow steppe. Precipitation had a negative effect in the desert steppe. Plant cover and plant density had positive effects in the typical steppe. In the meadow steppe, plant density had a marginally positive effect. No effect of vegetation parameters was found in the desert steppe. Finally, soil litter had a positive effect in the desert steppe. The use of the Shannon–Wiener index (Appendix A) and Shannon–Wiener exponential index (Appendix A) produced similar results.

#### 3.2.4. Dominance

Dominance (Simpson index, Table 4) was negatively affected by temperature in all three types of grasslands. Humidity had a negative effect in desert and typical steppes. Precipitation had a positive effect in the desert steppe and a negative effect in the meadow steppe. Plant biomass was positively related to dominance in the meadow steppe. Plant cover had a negative effect in the typical steppe. None of the soil factors were important to explain the species dominance except soil litter, which showed a negative effect in the desert steppe. The use of the Berger–Parker index gave similar outcomes (Appendix A). The use of the reciprocals of inverses of the Simpson and Berger–Parker indices produced nearly symmetrical results (Appendix A).

#### 3.2.5. Evenness

Evenness (Table 5) was positively influenced by humidity in the typical steppe. Precipitation had a negative effect in both the desert and the typical steppes and a positive effect in the meadow steppe. Soil temperature had a positive effect in the meadow steppe and soil litter in the desert steppe. Among the vegetation factors, plant biomass had a negative effect in the meadow steppe. Plant density and plant diversity were negatively related to *J* in the desert steppe.

## 4. Discussion

Several studies show that carabids respond to biotic and abiotic features of their local habitats [20,22,49,50,51,52,53,54,55,56]. Biotic factors like vegetation characteristics influence carabid community structure mainly through local effects. Since more vegetation sustains more arthropods in general, and carabids in particular, with more food resources [26,57], plant diversity and biomass influence carabid community [25,27,58]. Most carabids are active on the ground, some species lay eggs in burrows, and others overwinter as larvae or as adults in the soil. For these reasons, soil properties, such as softness, water content, temperature, etc., influence the carabid community [21,59,60,61,62]. Microclimatic conditions affect carabid assemblages, particularly through different species affinities for moisture, sunlight, and warmth. Temperature determines many physiological processes, because, in small ectothermic species, body temperature tends to depend directly on the ambient temperature [63,64]. Moreover, temperature preferences of a species vary according to the sexes of individuals and according to other climatic factors, such as precipitation [17], which may influence carabid abundance through its effects on vegetation [65,66]. Finally, humidity may also influence carabid abundance [17,67]. Our results indicate that the carabid community structure of the studied ecosystems responds to climatic, soil, and vegetation characteristics, but responses varied according to the community parameter under consideration (abundance, richness, diversity, dominance, and evenness) and the given ecosystem.

In a previous study, we found that climate factors influenced carabid functional diversity in different types of steppes [16]. Here, we found that climate factors were also important drivers of carabid activity density in the studied ecosystems. This result supports our hypothesis concerning the effect of climate factors, in particular that carabid activity–density increases with increasing temperature up to a threshold value, after which it decreases. It has been observed that, in fallow fields, the abundance of carabids depends on temperature [18]. Also, it is known that the thermoregulatory behavior may influence the response of the total activity density to temperature, because species activity can be stimulated by temperature [20]. This can explain the positive effect of temperature on the activity density and species richness in typical and meadow steppes. We found no significant effect in the desert steppe, possibly because, in this habitat, temperatures are always rather high. Moreover, higher temperatures stimulate activity, especially in predatory species, as they feed more at higher temperatures [19], but predatory carabids are less abundant in this habitat. Precipitation showed a positive effect on carabid activity density in the typical steppe, but a negative effect in the desert steppe and a non-significant effect in the meadow steppe. Precipitation also had a negative effect on richness in the desert steppe. These results reflect the particular environmental conditions of each type of steppe and are consistent with those of a companion study [16], which showed that rarefied species richness in the different types of grassland responded in the same way to climate factors. The response for habitat type was therefore similar to that for climate, notably temperature. Our finding of similar responses of activity density and richness suggests that the climatic factors that promote diversity also have positive effects on species abundances. Such results have been found in butterfly agroforest communities [68], where both abundance and species richness were significantly affected by rainfall and temperature. As regards humidity, it was positively related to species richness in the desert steppe and in the typical steppe, whereas no significant effect was found for the meadow steppe.

It is known that the amount of rainfall enhances the above-ground vegetation biomass in arid steppes [66,69]. Since vegetation provides carabids with food (directly for the herbivorous species and indirectly for the predatory ones) and shelter sites, more vegetation may allow carabids to reach higher densities. In the driest environment (i.e., the desert steppe), precipitation decreases the activity density of carabids, possibly because the communities of this habitat are mainly composed of species highly adapted to aridity. At intermediate conditions (i.e., the typical steppe), the effect is positive because here the community includes moderately hygrophilous species, which are locally more abundant/active where there is more rainfall; finally, in the less dry environment (i.e., the meadow steppe), there is no influence because precipitations are relatively abundant in all sampled sites.

Temperature and humidity effects on diversity were the same as for species richness, which suggests that species richness and species diversity respond similarly to certain environmental factors, and that the response of diversity (which encapsulates both richness and abundance) to climate factors is more driven by the richness component than the abundance component. We found a negative effect of temperature on dominance, whatever the type of grasslands, which indicates an inverse relationship between dominance and diversity. In the same way, humidity had negative effects on dominance in the desert and typical steppes. Precipitation increased the dominance in the desert steppe community, which suggests that precipitation promotes the activity of a few species, and depresses that of most of the community. By contrast, precipitation reduced the dominance in the meadow steppe.

As regards evenness, precipitation had a negative effect on species evenness in the desert steppe (thus paralleling the results for activity density and richness) and in the typical steppe (where, however, it acted positively on activity density), but a positive effect in the meadow steppe (where it does not have effect on activity density). Overall, these results indicate that in the desert steppe, precipitation reduces both the number of species and their abundances, possibly because the carabids of this environment are strongly adapted to arid conditions; by contrast, in the typical steppe, precipitation increases abundances, but this increase was not even among species, leading some species to become more dominant, with a negative effect on the evenness values. In the meadow steppe, communities inhabiting sites with higher precipitation seem to have a higher evenness, because precipitation does not increase overall abundance in this ecosystem. Humidity was negatively related to species evenness in the typical steppe, where it had a positive effect on richness. We can postulate that an increase in humidity promotes the dominance of certain species (the hygrophilous ones) and negatively affects others (the xerophilous ones), leading to a decrease in evenness. Overall, our results indicate that the influence of climatic factors on community structure is complex, and varies according to the habitat and the aspect of structure that is being considered.

The effects of soil characteristics on carabid abundance patterns have been investigated in many studies [62,70,71]. We found that soil factors were weak predictors of the activity density of carabids in steppes, but soil bulk density was negatively related to species richness in the typical steppe, as also observed using rarefied richness values [16]. Soil bulk density is an indicator of soil compaction, which has negative effects on egg laying or adult burrowing for diapause [21], an aspect that should be tested in the future. However, soil bulk density did not significantly affect species richness in the meadow and in the desert steppe. In the desert steppe, soil compaction may reflect the presence of vegetation spots, which may attract carabids, thus counteracting the negative effects reported above. In fact, even a positive effect of soil bulk density on carabid rarefied richness was previously found in this environment [16]. As regards the influence of soil factors on diversity and dominance, we found that soil litter positively influenced diversity and evenness, and negatively influenced dominance, in the desert steppe. Carabids are known to be influenced by the amount of soil litter, because it modulates both soil moisture and temperature, and improves fertility, food availability, and habitat heterogeneity [21,22,23,61]. These important functions are further enhanced in desert ecosystems, where vegetable detritus represents a prominent trophic source. Thus, soil litter may be an important driver of carabid diversity in the desert steppe more than in other habitat types by providing these insects with food, water, shelter, lower temperatures, and higher humidity. We also found that soil temperature had a positive effect on evenness in the meadow steppe, possibly reducing the activity of the most dominant species.

Except for a positive effect of biomass in the meadow steppe, we recorded no effect of vegetation factors on activity density. This is in contrast with our prediction, based on the positive influence of vegetation on carabid activity density observed in other contexts [72]. However, this lack of a shrub effect might be due to the fact that species can change their preference for the shrubs during the year. For example, in a sandy desert in the central parts of the Heihe River basin (north-western China), the activity density of ground-dwelling beetles was higher under shrubs than in bare habitats in spring, but the reverse occurred in autumn [73].

In general, vegetation factors did not affect richness but had strong effects on diversity. Plant cover and density had positive effects on carabid diversity in the typical steppe and meadow steppe but not in the desert steppe. These positive relationships found in our study contrast with a study finding that herb density was not important to predict carabid diversity in a forest ecosystem [27]. This inconsistency may be due to differences in the ecosystems. Vegetation cover might promote carabid diversity by creating a heterogeneous and stratified microenvironment supporting different species [74]. While this role can be very important in environments dominated by herbs, such as the typical and meadow steppes, it may be less relevant in the desert steppe (where vegetation is too sparse and its structure too simple and homogeneous to have detectable influences on carabid diversity) or in forests (where the vegetation structure is dominated by trees, not by herbs). Plant biomass increased dominance and decreased evenness in the meadow steppe (where plant biomass was found to have a negative effect on functional diversity based on traits related to movement capabilities [16]), whereas plant cover decreased dominance in the typical steppe (in accordance with the positive influence on diversity). Also, plant density and plant species richness had a negative effect on evenness in the desert steppe, which is again consistent with the negative influence of plant density on functional diversity based on traits related to movement capabilities [16]. Overall, these negative effects of vegetation characteristics mean that vegetation tends to promote the dominance of certain species over rare species, thus reducing the evenness.

To our knowledge, this is the first study that investigated the impact of local habitat characteristics on carabid evenness in steppe ecosystems. Some authors claim that evenness should be positively correlated with richness [75,76], but the opposite can also be true, if species richness increases by the addition of rare species (thus decreasing evenness). In fact, empirical evidence suggests that relationships between richness and evenness can be positive, negative, or absent, depending on the study system [77]. Our study supports that evenness and species richness respond differently to local drivers and that the response depends on the type of grassland.

Our expectation that abiotic conditions (such as climate, soil compaction, and soil temperature) were more important in determining carabid community structure in the desert steppe, whereas biotic conditions (such as vegetation characteristics) were more important in the typical and meadow steppe was confirmed. In fact, we found that climatic factors were the most important characteristics influencing carabid community structure in all three types of steppe, but vegetation characteristics also had significant effects in the typical and meadow steppe. This indicates that, in accordance with theory [78], biotic factors become important only when environmental severity decreases.

## 5. Conclusions

Carabid communities in steppe ecosystems respond to environmental factors in different ways according to the type of steppe. In general, climate factors were the most important drivers of community structure, whereas the effects of soil and vegetation factors were weak. The desert steppe was the ecosystem with the lowest carabid activity density, species richness, diversity, and evenness and with the highest dominance. This indicates that the carabid community of this ecosystem is relatively simple and highly dominated by a few species that are best adapted to cope with the severe conditions of this environment. The typical and meadow steppes present similar carabid communities characterized by a more complex structure. As expected on the basis of environmental severity, the most severe ecosystem (i.e., the desert steppe) was only influenced by climatic factors, whereas a certain influence of biotic factors emerged in the other, less severe, ecosystems.

## Figures and Tables

**Table 1 insects-11-00019-t001:** Results of RE-ESF analysis (Random Effect Eigenvector Spatial Filtering) between habitat characteristics and carabid activity–density for the three grassland types.

Variables		Grassland Type		
		Desert Steppe	Typical Steppe	Meadow Steppe
r^2^		0.17	0.36	0.42
Vegetation	PB	1.24 ± 1.08 (0.248)	0.32 ± 0.79 (0.690)	**2.71 ± 1.20 (0.026)**
	PC	2.58 ± 1.51 (0.099)	−0.28 ± 0.85 (0.742)	2.12 ± 1.11 (0.060)
	PD	−3.39 ± 1.99 (0.099)	0.35 ± 0.81 (0.664)	0.26 ± 1.14 (0.821)
	PH	−1.38 ± 1.57 (0.386)	−0.21 ± 0.84 (0.800)	−3.60 ± 1.85 (0.053)
	PSD	1.32 ± 1.62 (0.421)	−0.60 ± 0.58 (0.305)	−0.09 ± 1.07 (0.933)
Soil	SBD	0.22 ± 1.18 (0.853)	0.31 ± 0.57 (0.593)	0.22 ± 1.25 (0.861)
	SL	0.54 ± 1.18 (0.650)	−0.61 ± 0.87 (0.484)	−0.67 ± 0.97 (0.488)
	SM	−0.74 ± 1.22 (0.547)	0.01 ± 0.54 (0.989)	−0.94 ± 1.21(0.442)
	ST	----	−0.68 ± 0.58 (0.240)	−1.72 ± 1.68 (0.308)
Climate	Hum	2.17 ± 1.72 (0.218)	1.17 ± 0.92 (0.202)	2.45 ± 1.62 (0.132)
	Prec	**−3.80 ± 1.63 (0.027)**	**3.76 ± 0.67 (<0.001)**	1.45 ± 1.44 (0.318)
	Temp	2.27 ± 1.92 (0.249)	**2.30 ± 0.77 (0.003)**	**6.75 ± 1.16 (<0.001)**
	Intercept	**8.24 ± 0.92 (<0.001)**	**6.74 ± 0.45 (<0.001)**	**8.75 ± 0.83 (<0.001)**

r^2^ = adjusted coefficient of determination. Parameter estimated coefficients (±standard error) and *p*-values (in parentheses) are given for each predictor. Significant effects are in bold. Predictors abbreviations: PB: Plant dry biomass, PC: Plant cover, PD: Plant density, PH: Plant height, PSD: Plant species diversity (richness); SBD: Soil bulk density, SL: Soil litter, SM: Soil moisture, ST: Soil temperature; Hum: Humidity, Prec: Precipitation, Temp: Temperature.

**Table 2 insects-11-00019-t002:** Results of RE-ESF analysis between habitat characteristics and carabid richness (Chao 1) for the three grassland types.

Variables		Grassland Type		
		Desert Steppe	Typical Steppe	Meadow Steppe
r^2^		0.56	0.29	0.27
Vegetation	PB	0.08 ± 0.14 (0.582)	0.25 ± 0.25 (0.324)	0.06 ± 0.27 (0.823)
	PC	0.16 ± 0.20 (0.444)	0.41 ± 0.27 (0.137)	−0.13 ± 0.25 (0.600)
	PD	−0.39 ± 0.27 (0.159)	0.30 ± 0.25 (0.227)	0.12 ± 0.25 (0.636)
	PH	−0.17 ± 0.21 (0.443)	−0.35 ± 0.27 (0.197)	−0.19 ± 0.37 (0.607)
	PSD	−0.44 ± 0.23 (0.062)	0.03 ± 0.19 (0.867)	0.20 ± 0.24 (0.420)
Soil	SBD	0.08 ± 0.16 (0.628)	**−0.38 ± 0.18 (0.036)**	−0.40 ± 0.29 (0.174)
	SL	0.29 ± 0.17 (0.102)	−0.20 ± 0.28 (0.484)	0.26 ± 0.24 (0.271)
	SM	0.30 ± 0.16 (0.077)	−0.01 ± 0.17 (0.934)	−0.01 ± 0.27 (0.960)
	ST	----	**−0.40 ± 0.18 (0.028)**	0.27 ± 0.37 (0.464)
Climate	Hum	**0.78 ± 0.22 (0.002)**	**1.09 ± 0.29 (<0.001)**	0.25 ± 0.34 (0.463)
	Prec	**−0.87 ± 0.22 (<0.001)**	0.28 ± 0.21 (0.180)	0.08 ± 0.33 (0.813)
	Temp	0.39 ± 0.25 (0.844)	**1.16 ± 0.25 (<0.001)**	**1.22 ± 0.27 (<0.001)**
	Intercept	**1.85 ± 0.12 (<0.001)**	**4.00 ± 0.14 (<0.001)**	**4.18 ± 0.19 (<0.001)**

r^2^ = adjusted coefficient of determination. Parameter estimated coefficients (± standard error) and *p*-values (in parentheses) are given for each predictor. Significant effects are in bold. Predictors abbreviations as in Table 1.

**Table 3 insects-11-00019-t003:** Results of the RE-ESF analysis between habitat characteristics and carabid diversity (Brillouin index) for the three grassland types.

Variables		Grassland Type		
		Desert Steppe	Typical Steppe	Meadow Steppe
r^2^		0.34	0.38	0.47
Vegetation	PB	0.05 ± 0.04 (0.241)	0.04 ± 0.04 (0.247)	−0.04 ± 0.03 (0.242)
	PC	0.05 ± 0.06 (0.397)	**0.10 ± 0.04 (0.006)**	−0.01 ± 0.03 (0.748)
	PD	−0.16 ± 0.08 (0.056)	**0.09 ± 0.03 (0.003)**	**0.07 ± 0.03 (0.043)**
	PH	−0.02 ± 0.06 (0.777)	−0.00 ± 0.04 (0.992)	0.01 ± 0.06 (0.904)
	PSD	−0.13 ± 0.07 (0.061)	−0.03 ± 0.03 (0.267)	0.03 ± 0.03 (0.362)
Soil	SBD	−0.01 ± 0.05 (0.896)	−0.03 ± 0.03 (0.185)	−0.00 ± 0.04 (0.989)
	SL	**0.11 ± 0.05 (0.025)**	−0.04 ± 0.04 (0.377)	−0.00 ± 0.03 (0.934)
	SM	0. 08 ± 0.05 (0.091)	−0.01 ± 0.03 (0.731)	0.01 ± 0.03 (0.677)
	ST	----	−0.05 ± 0.03 (0.096)	0.09 ± 0.05 (0.052)
Climate	Hum	**0.17 ± 0.06 (0.012)**	**0.17 ± 0.04 (<0.001)**	−0.03 ± 0.05 (0.517)
	Prec	**−0.23 ± 0.07 (0.001)**	0.05 ± 0.03 (0.123)	0.08 ± 0.04 (0.066)
	Temp	0.14 ± 0.07 (0.055)	**0.17 ± 0.04 (<0.001)**	**0.19 ± 0.03 (<0.001)**
	Intercept	**0.22 ± 0.04 (<0.001)**	**0.65 ± 0.02 (<0.001)**	**0.65 ± 0.02 (<0.001)**

r^2^ = adjusted coefficient of determination. Parameter estimated coefficients (± standard error) and *p*-values (in parentheses) are given for each predictor. Significant effects are in bold. Predictors abbreviations as in Table 1.

**Table 4 insects-11-00019-t004:** Results of RE-ESF analysis between habitat characteristics and carabid dominance (Simpson index) for the three grassland types.

Variables		Grassland Type		
		Desert Steppe	Typical Steppe	Meadow Steppe
r^2^		0.29	0.23	0.28
Vegetation	PB	−0.05 ± 0.03 (0.119)	−0.01 ± 0.02 (0.643)	**0.06 ± 0.02(0.012)**
	PC	−0.03 ± 0.04 (0.492)	**−0.05 ± 0.02 (0.026)**	0.01 ± 0.02 (0.788)
	PD	0.13 ± 0.06 (0.051)	−0.03 ± 0.02 (0.122)	−0.03 ± 0.02 (0.163)
	PH	0.00 ± 0.04 (0.926)	−0.01 ± 0.03 (0.706)	−0.01 ± 0.03 (0.780)
	PSD	0.10 ± 0.05 (0.073)	0.02 ± 0.02 (0.383)	−0.01 ± 0.02 (0.665)
Soil	SBD	−0.01 ± 0.04 (0.758)	0.02 ± 0.02 (0.214)	−0.01 ± 0.03 (0.843)
	SL	**−0.09 ± 0.04 (0.023**)	0.04 ± 0.03 (0.153)	−0.01 ± 0.02 (0.789)
	SM	−0.07 ± 0.04 (0.071)	0.01 ± 0.02 (0.548)	−0.01 ± 0.02 (0.696)
	ST	----	0.01 ± 0.02 (0.551)	−0.06 ± 0.03(0.058)
Climate	Hum	**−0.11 ± 0.05 (0.034)**	**−0.01 ± 0.03 (<0.001**)	0.02 ± 0.03 (0.459)
	Prec	**0.17 ± 0.05 (0.003)**	0.01 ± 0.02 (0.541)	**−0.06 ± 0.03 (0.042)**
	Temp	**−0.12 ± 0.06 (0.047)**	**−0.07 ± 0.02 (0.003)**	**−0.07 ± 0.02 (0.007)**
	Intercept	**0.82 ± 0.03 (<0.001)**	**0.53 ± 0.01 (<0.001)**	**0.50 ± 0.02 (<0.001)**

r^2^ = adjusted coefficient of determination. Parameter estimated coefficients (± standard error) and *p*-values (in parentheses) are given for each predictor. Significant effects are in bold. Predictors abbreviations as in Table 1.

**Table 5 insects-11-00019-t005:** Results of RE-ESF analysis between habitat characteristics and carabid evenness (Pielou index) for the three grassland types.

Variables		Grassland Type		
		Desert Steppe	Typical Steppe	Meadow Steppe
r^2^		0.28	0.07	0.11
Vegetation	PB	0.10 ± 0.05 (0.064)	0.00 ± 0.04 (0.994)	**−0.10 ± 0.04 (0.006)**
	PC	0.05 ± 0.07 (0.417)	0.07 ± 0.03 (0.052)	0.00 ± 0.03 (0.929)
	PD	**−0.21 ± 0.10 (0.048)**	−0.00 ± 0.03 (0.985)	0.03 ± 0.03 (0.375)
	PH	0.04 ± 0.07 (0.569)	0.01 ± 0.04 (0.700)	0.02 ± 0.04 (0.604)
	PSD	**−0.18 ± 0.08 (0.037)**	−0.01 ± 0.03 (0.831)	−0.00 ± 0.03 (0.907)
Soil	SBD	0.02 ± 0.06 (0.695)	−0.01 ± 0.02 (0.601)	0.03 ± 0.04 (0.419)
	SL	**0.16 ± 0.06 (0.016)**	−0.06 ± 0.04 (0.095)	−0.00 ± 0.03 (0.936)
	SM	0.09 ± 0.06 (0.118)	−0.00 ± 0.02 (0.850)	0.02 ± 0.04 (0.512)
	ST	----	0.02 ± 0.03 (0.492)	**0.10 ± 0.05 (0.031)**
Climate	Hum	0.15 ± 0.08 (0.071)	**0.11 ± 0.04 (0.007)**	−0.05 ± 0.04 (0.228)
	Prec	**−0.17 ± 0.08 (0.005)**	**−0.06 ± 0.03 (0.048)**	**0.09 ± 0.04 (0.042)**
	Temp	0.17 ± 0.09 (0.064)	0.05 ± 0.03 (0.189)	0.01 ± 0.04 (0.798)
	Intercept	**0.31 ± 0.05 (<0.001)**	**0.70 ± 0.02 (<0.001)**	**0.73 ± 0.03 (<0.001)**

r^2^ = adjusted coefficient of determination. Parameter estimated coefficients (± standard error) and *p*-values (in parentheses) are given for each predictor. Significant effects are in bold. Predictors abbreviations as in Table 1.

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
