# Peer review of "Influence of Climate and Local Habitat Characteristics on Carabid Beetle Abundance and Diversity in Northern Chinese Steppes"

_insects, 2019, doi:10.3390/insects11010019_

Round 1

Reviewer 1 Report

the revised version of the manuscript has been improved according to my suggestions

Author Response

Dear Reviewer,

Thank you for appreciating our revision. We are happy you are satisfied with this new version.

All the best,

Noelline Tsafack

(on behalf of all coauthors).

Reviewer 2 Report

The authors did a fine job responding to my suggestions.  I think the discussion flows much better.  I've suggested a few more stylistic changes on the manuscript that the authors can consider.

Author Response

Dear Reviewer,

Thank you very much for giving us the possibility of submitting an improved version of our manuscript entitled: “Influence of climate and local habitat characteristics on carabid beetle abundance and diversity in northern Chinese steppes” submitted to the journal Insects.

We are very grateful for these further comments.

We incorporated all corrections and suggestions.

The authors did a fine job responding to my suggestions.  I think the discussion flows much better.  I've suggested a few more stylistic changes on the manuscript that the authors can consider.

Authors: Thank you for appreciating our revision. We are happy you are satisfied with this new version. We have implemented your further corrections.

Page 11: Line 347-348: “This result is not surprising, given the inverse conceptual relationship between dominance and diversity, although the two indices used here are not mathematically linked.”

You say this several times, but I'm not convinced. Dominance is the relative abundance (p = n/N) of the most abundant species. This same value is used in all diversity indices. The contributions of the other species are present in those indices, too, but this very same value of the most abundant species factors into both computations.  So, it seems they are 'linked' by this value, at least.

Authors: We have rephrased this sentence in the following way: "We found a negative effect of temperature on dominance, whatever the type of grasslands, which indicates an inverse relationship between dominance and diversity."

Page 12: Line 372: Change “affect significantly” to “significantly affect”

Authors: Done.

Page 12: Line 376: Change “influenced positively” to “positively influenced”

Authors: Done.

Page 12: Line 377: change “negatively dominance” with " influenced negatively dominance”

Authors: Done.

Page 13: Line 410-411: this begs a citation for this 'claim'. I'd suggest that the opposite is more likely, as species richness usually increases as a consequence of adding rare species (thus decreasing evenness).

Authors: We agree, and we added two references and reworked the sentence.

Page 13: Line 420: Menge and Sutherland 76? 78? would be a good reference to cite here on this point, rather than a general text.

Authors: we added the citation.

Thanking this reviewer for his/her helpful comments,

All the best,

Noelline Tsafack

(on behalf of all coauthors).

Reviewer 3 Report

The authors have responded well to my minor comments. However, my main concern regarding this study was the weak approach to the analyses, which is simply based on a battery of indices. Whilst the authors state in their covering letter that they have addressed this issue, they still include no less than seven indices. Such indices can be used to compare diversity between a set of sites but are not adequate as the only analysis method. The study design is sound, the data seems to be sound, and the study system is very novel and interesting. However, the authors present a manuscript in which all the attention is focused on comparing these indices, rather than a study of the novel assemblages of a little known habitat system. I would encourage them to rethink their strategy for analyzing this data, think about their objectives for the study and the manuscript and then develop their approach on the basis of their objectives.  

Author Response

Dear Reviewer,

Thank you very much for giving us the possibility of submitting an improved version of our manuscript entitled: “Influence of climate and local habitat characteristics on carabid beetle abundance and diversity in northern Chinese steppes” submitted to the journal Insects.

We are very grateful for these further comments.

We incorporated all corrections and suggestions.

The authors have responded well to my minor comments.

Authors: Thank you for appreciating our efforts.

However, my main concern regarding this study was the weak approach to the analyses, which is simply based on a battery of indices.

Authors: One purpose of the study was to determine which indices were best suited to describe different characteristics (diversity, dominance, evenness) of steppe carabid communities and the second was to investigate the drivers of these characteristics.

Thus, to determine the most suitable indices, we have to compare the “battery of indices”, since no single index can adequately describe all aspects of community structure (This is explained in the text line 82). We cannot understand how this approach can be considered weak. This approach is well consolidated in community ecology research. Using diversity indices and relating them to environmental variables is a straightforward way to analyse how community structure is influenced by the environment.

Whilst the authors state in their covering letter that they have addressed this issue, they still include no less than seven indices. Such indices can be used to compare diversity between a set of sites but are not adequate as the only analysis method.

Authors: We found this comment rather vague and we cannot find a clear indication of what the referee really suggests. We included in the main text results about four indices measuring respectively species richness (Chao), species diversity (Brillouin index), species dominance (Simpson index) and species evenness (Pielou index) and the other indices were presented in the supplementary material, including the results about Hill number previously suggested by the reviewer in the round3. So, we are a bit embarrassed, since this reviewer provided contrasting comments and his/her comments are in contrast with those of the other reviewers. We think that our choice of selecting a minimum set of indices to express different community aspects represents the best way to integrate his/her comments with those of the other reviewers.

The study design is sound, the data seems to be sound, and the study system is very novel and interesting.

Authors: We thank the reviewer for this positive comment.

However, the authors present a manuscript in which all the attention is focused on comparing these indices, rather than a study of the novel assemblages of a little known habitat system. I would encourage them to rethink their strategy for analyzing this data, think about their objectives for the study and the manuscript and then develop their approach on the basis of their objectives.  

Authors: We are sorry, but we really cannot understand this comment, which is vague and in contrast with those of the other reviewers. Our objectives are clearly indicated, point by point, at the end of the Introduction and the use of different indices are the way to address these objectives.

We really hope that you are satisfied with this last revision.

All the best,

Noelline Tsafack

(on behalf of all coauthors).

This manuscript is a resubmission of an earlier submission. The following is a list of the peer review reports and author responses from that submission.

Round 1

Reviewer 1 Report

I see the paper worth to be published after some minor changes

line 54 - there is also the paper of Talarico et al 2016 on seed-feeding carabids

138 - sampling that using less --> sampling than using less

178 - "we think that..." is a little bit weak, I suggest "it is likely that..."

195 - "or that are might be..."

200 - this definition holds true if you compute Simpson's index of Diversity as 1-D. Is it a typo or is it a mistake?

206 - following your definition of Simpson Diversity, this holds true only for d

354 - is this a case where the Intermediate Disturbance Hypothesis is confirmed?

375 - you should write if it is worth of further investigation (i.e. data lacking), otherwise give an explanation.

391 - do you know the egg-laying behaviour of your species? is this a case of information lacking needing further laboratory experiments?

432 - too vague, at least give a temptative analysis of the vagility of the most abundant species

there are some points that seems to be counter-intuitive, e.g. lines 341-344, 365-366, 397-399

you should give more clear explanation about the fact that they, on the contrary, support your hypotheses

no species appear in the discussion and conclusion, while there are several references to "certain species", it seems that there is poor knowledge of species ecology

Author Response

Dear Reviewer,

Thank you very much for giving us the possibility of submitting an improved version of our manuscript entitled: “Influence of climate and local habitat characteristics on carabid beetle abundance and diversity in northern Chinese steppes” submitted to the journal Insects.

We are very grateful to the reviewer for her/his comments.

We did our best to incorporate all her/his corrections and suggestions.

Please see the attachment for the response to your comments point by point and for the revised version of the manuscript.

We would like to thank once again the reviewer for her/his comments, which helped us to ameliorate the manuscript. We hope that you are satisfied with this revision and that the manuscript is now in a suitable form for publication.

All the best,

Noelline Tsafack

(on behalf of all coauthors)

Reviewer 2 Report

The paper by Tsafack, et al. provides an excellent survey of the relative importance of habitat parameters in determining the structure of carabid beetle diversity in three habitats in the Chinese steppes.  I think the methodology is sound, and the analyses are appropriate, however I have several questions regarding their presentation and several suggestions for how the delivery of the results and discussion could be improved.

This is a tour-de-force of statistical hypothesis testing, examining the effects of 12 habitat parameters on 8 community descriptors across three ecosystems examined separately and together. Obviously, the dimensionality of this analysis is huge, and describing all of the effects is a huge task that will necessarily produce a lot of redundancy.  I understand that the authors include so many community descriptors in an effort to determine which descriptors are most sensitive to which parameters, but I find that the inherent correlations among community indices and the redundancies in the discussion make conclusions hard to follow.  Perhaps the flow from results through conclusions would be easier to understand if some of the redundancies were removed.  For example, both Chao-1 and Margalef indices of richness vary the same way across the habitats; both the Brillouin and Shannon-Weiner vary the same way, as do the three diversity indices (Pielou, Simpsons, Berger-Parker dominance).  The mathematical similarities among these indices are interdependent, and guarantee that patterns across large datasets like this will be the same.  The complexity of the discussion, in which all indices are discussed, could be greatly reduced by eliminating redundant indices.  The number of independent variables is daunting enough; eliminating redundant dependent variables would simplify the results and discussion tremendously, making arguments much clearer with no loss in explanatory value.

Also, in the interests of reducing the dimensionality, I wonder as to the necessity of the ‘regional scale’ patterns.  These are simply the effects across all the habitats, combined, but it is really the differences BETWEEN habitats that are the focus of this study.  The fact that, in general, increasing precipitation correlates with increasing abundance is not really important.  The real pattern of interest is that it has a positive effect in typical and meadow steppe but a negative effect in desert steppe. There are only two circumstances where there is a regional pattern BUT NOT an effect in one habitat or another (Humidity vs. abundance, PB vs. diversity).  As such, all other variables are discussed at the habitat level, anyway, so gross patterns at the regional scale are redundant consequences of patterns at the habitat scale.  The two others, listed above, may be significant because of the combined sample size and consistency of pattern across habitats, but little is lost in excluding these patterns because the goal, again, is to describe how carabid communities differ between these habitats.

The discussion is unavoidably complex.  There are two pathways possible – deal with each dependent variable (abundance, richness, and diversity) and describe which factors affect each variable.  This is the pathway the authors take.  However, the other pathway would be to consider classes of independent variables, and summarize their effects on all the dependent variables together. Given the correlations between abundance, richness, and diversity, this might be the most streamlined and efficient way through all the contrasts and reduce redundancy. The authors begin this way, discussing the overarching effects of humidity and the weak contribution of soil effects, but then they shift gears and consider each dependent variable separately. However, since all three metrics are trying to describe some element of community structure, I think it is more natural to look at a set of independent variables and see how they affect community structure (as reflected by the three indices).

It would also be natural to consider each habitat separately.  So, the discussion could begin with the climatic parameters… something like this:

Climatic parameters had different effects on carabid community structure in different steppe environments. In the desert steppe, precipitation had a negative effect on abundance, richness, and diversity, while humidity had positive effects on richness and diversity. In contrast, in typical and meadow steep, temperature was the most important climatic driver of carabid community structure, with abundance, richness, and diversity increasing with increased temperature. Then, present the relevant info for temp and precip effects as in the discussion.

Then move to vegetational effects in each habitat

Then to soil effects in each habitat.

I think this will reduce the redundancy and make the discussion smoother and the conclusions and patterns more obvious.

In another vein, I wonder why the authors include three sets of model parameters (rloglnk, AIC, BIC). To which of these modeling parameterization options do the r2 and variable parameterizations refer? Each parameterization option would give a different fit, with different parameter coefficients, I believe. It seems that the entire R output was transferred to tables.  As mentioned, the portion of Table 1 that presents mean comparison information is entirely redundant to figure 1 and should be deleted.  Again, it seems all R output was simply transposed without the necessary digestion.

I have made several editing suggestions on the ms.

Author Response

Dear Reviewer,

Thank you very much for giving us the possibility of submitting an improved version of our manuscript entitled: “Influence of climate and local habitat characteristics on carabid beetle abundance and diversity in northern Chinese steppes” submitted to the journal Insects.

We are very grateful to the reviewer for her/his very detailed and useful comments.

We did our best to incorporate all her/his corrections and suggestions.

In the following, we explain how we dealt with each comment.

Please see the attachment for the response to your comments point by point and for the revised version of the manuscript.

We would like to thank once again the reviewer for her/his very detailed and useful comments, which helped us to ameliorate the manuscript. We hope that you are satisfied with this revision and that the manuscript is now in a suitable form for publication.

All the best,

Noelline Tsafack

(on behalf of all coauthors)

Reviewer 3 Report

Please refer to attached document

Author Response

(The authors gave the same response as above.)

Round 2

Reviewer 3 Report

Please refer to my comments in the attached file

Author Response

 Dear Reviewer,

Thank you very much for giving us the possibility of submitting an improved version of our manuscript entitled: “Influence of climate and local habitat characteristics on carabid beetle abundance and diversity in northern Chinese steppes” submitted to the journal Insects.

We are very grateful for these further comments.

We did our best to incorporate all her/his corrections and suggestions.

In the following, we explain how we dealt with each comment.

The authors appear to have implemented virtually all of the recommended minor comments and made a good revision of their manuscript, which is much improved. However, the improvement regarding the major comments is more moderate. The main problem with this manuscript is that the Discussion does not fit with the first part of the manuscript. The abstract gives a clear description of what the authors set out to study ‘the effects of fine-scale factors (soil and vegetation) and coarse-scale factors (climate) on carabid community parameters (abundance, richness, diversity, dominance, and evenness) in three types of steppes (desert, typical, and meadow steppes) in northern China’ and goes on to summarise the results. The study summarised in the abstract is very good and very interesting. However, the Results and Discussion sections are dominated by a large battery of diversity indices. The manuscript would be greatly improved by removing most of the material on these indices and focus on the analyses that provide results relevant to the research questions.

Authors: Thank you for your encouraging comments. We have removed most of the indices. Results and Discussion are now focused on only one index of diversity, one of dominance and one of evenness. In accordance with best practices and literature stressing the need of considering all these three aspects (diversity, dominance and evenness), we think that all these three aspecets must be presented. We have clarified in the introduction how these three aspects are relevant to the research questions.

Also there is still very little mention of relevant scientific theory, such as functional diversity, disturbance, intermediate disturbance hypothesis, habitat degradation and climate change, for instance. They don’t have to all be covered but the manuscript would be greatly improved by reference to one or two relevant theories, and introducing and justifying them in the Introduction and then using them to explain the results in the Discussion. The Discussion has improved, though the opening paragraph focuses rather much on justifying the study and includes some speculation.

Authors: We have expanded the introduction and reformulated the various point explaining the expected patterns. Moreover, we have introduced two new paragraphs and we explained how our reserach refers to the theory regarding the different importance of abiotic and biotic factors in determining community structure according to environmental severuity. We referred to this theory in the Discussion and in the final new sentence of the Conclusions. We have shortened the openining paragraph of the Discussion and deleted speculations.

There is still some basic information missing from the Materials and Methods section. The sectors, mentioned in lines 100-105, should be explained in the methods section somewhere. How were they defined and what was their purpose? Presumably the authors expected that there would be some difference between the results of different sectors? Was there a research question about them?

Authors:  We have now expalined that, to reflect a possibly within-ecosystem variability, we placed traps into sectors identified by the main vegetation type.

Information is also missing on precisely how the pitfall traps were arranged. Were they in a transect or a grid?

Authors: Pitfall traps were placed randomly with no particular arrangment. We have added this information in the text.

How far apart from each other?

Authors: Traps were separated by at least five meters from each other. This is indicated in the text.

What size of trap?

Authors: The size of the trap was: diameter: 7.15 cm, depth: 9 cm. This is indicated in the text.

What was the preservative solution used in them?

Authors: Alcohol. This is indicated in the text.

Were lids used to protect them?

Authors: Pitfalls traps were covered by a lid. We have now specified this point.

For future reference please note that it is generally not a good idea to pool samples. Keeping the samples obtained at different traps at the same site separate gives finer information for the analyses and sensitivity is lost if they are pooled.

Authors: Thank you for this suggetion for further research.

Also there is no information on whether any traps were lost and how these were compensated for in the analyses.

Authors: No traps were lost. We have added this information now.

The authors state that some of the sites were very close to each other (line 134). It is also relevant to provide information in the Materials and Methods section on the size of the sites and how close to the borders between different sites were the sampling points. This information is important so that readers can consider how representative the sampling points were, or is there a risk that some sampling points would trap species of adjacent habitats.

Authors: The distance between the sampling areas in the three ecosystems (desert, typical and meadow steppes) were as follows: the desert was about 250 km from the meadow and the typical steppe; the typical and the meadow steppe were about 70 km apart. Thus, there is no risk that that some sampling points would trap species of adjacent habitats. Sampling sites were separated by at least 150 m from each other to minimize spatial autocorrelation. We have provided this information in the text.

The approach to the analyses is very weak. The authors use R, which suggests that they have the capacity for producing good statistical analyses. However, the analyses are primarily based on seven different indices, which are basically seven ways of comparing the numbers of species and numbers of individuals. It would be much more effective to use just one analysis of the data, aimed at addressing the research questions. Readers are going to be much more interested in an analysis of carabid assemblages of three different steppe habitats under different disturbance regimes, as described in the abstract, than an unnecessary comparison of diversity indices. Basically four diversity indices generate the same results (Figs 1b-1e), the two dominance indices give the same result (Fig. 1f-1g) and the evenness plot gives a contrasting result to the dominance indices. The best approach to analysing this data would be to use a GLM or alternatively such a method as HMSC-R, for instance. If the authors wish to use indices, then unless there is a well-justified reason for using more than one, I recommend to use just one diversity index and one dominance index, and delete the rest of the indices from the manuscript. This would generate a more concise manuscript that is more focused on the findings. Currently, much of this manuscript is taken up with comparing indices, whereas readers are likely to be much more interested to learn about the carabid assemblages of the studied habitats and the effects of the environmental factors considered.

Authors: We have now deleted all redundant indices from the main text. Now, the manuscript presents results for only one index of diversity, one of dominance and one of evenness.

We did not use GLM because of the possible autocorrelation of the sampling. Thank you for the suggestion for the new method HMSC (Hierarchical Modelling of Species Communities). We hope to use it in the future; however, we are interesting here in community structure parameters as expressed by indices. We have better explained this point in the Introduction.

Some sort of community level analysis, such as an NMDS ordination, would also have been valuable, to compare the assemblages of the compared habitats.

Authors: We compared assembalges in another study [Reference # 16], where we used Canonical correspondence analyses (CCAs) and cluster analyses. Incidentally, we already performed a NMDS for the previous study, but resusts were similar to those obtained with the CCAs.

The Discussion has been extensively revised and it is much improved.

Authors: Thank you for your encouraging comments. We followed the recommendations of the three reviewers.

The first paragraph now contains a lot of very relevant information about carabids, though there is also some rather speculative information too, as indicated in my minor comments. It feels rather as if too much information is squeezed into this one paragraph but it is ok. It would have been better to have just given a general, lighter introduction to factors relevant to carabid assemblages of steppe and arid habitats, and save some of the more specific information for later in the Discussion.

Authors: We have reduced this part of the Dicsussion and deleted speculative information.

The Conclusions are appropriate and seem fine, though it is not clear to me on what analyses the conclusions regarding human disturbance and climate change are based. Currently the Discussion is the part of this manuscript that requires most work, and it should be made to focus on the research questions, be based on the results of the analyses and lead clearly to the conclusions. Hopefully that will all be clear when the Discussion has been revised again. At the moment is not at all clear what the climate conclusions and human disturbance conclusions are based on.

Authors: We have deleted climate change and human disturbance from conclusions. We have modified the Discussion and Conclusions to make them more consistent with the Results and the questions presented in the Introduction.

Minor comments

line 26 because highly productive grasslands are being transformeding into arid lands.

Authors: We have deleted this sentence because we have removed any speculation about climate change and human disturbance.

line 33 40% of the China’s land surface,

Authors: We have deleted the word.

linie 42 sensitive to the habitat alteration,

Authors: We have deleted the word.

line 54 diversity at the regional scale.

Authors: We have corrected the sentence.

line 56 on various aspects of carabid

Authors: We have added the plural.

line 59 influences of fine- and coarse-scale factors (deletion of hyphen after fine)

Authors: We have deleted the hyphen after fine.

line 80 relatively few species, strongly adapted

Authors: We have added the coma.

line 81 hypothesize, therefore, that deserts will have communitiesy with strong dominance,

Authors: We have corrected the sentence.

line 90 tolerant species, like such as Agropyron mongolicum,

Authors: We have corrected the sentence.

line 119-120 pitfall traps were put downset once a month in mid-month,

Authors: We have corrected the sentence.

line 120 prior to collectionemptying and removal of the traps.

Authors: We have corrected the sentence.

line 125 populations were not over-captured due to frequent samplingover-sampled.

Authors: We have corrected the sentence.

lines 230-231 None of the soil or vegetation factors was important to explainhad a significant effect on activity density,

Authors: We have corrected the sentence.

line 296 It isn’t very relevant to talk about indicators here, as they are not used as indicators in this study. Such justification is not really necessary.

Authors: We have deleted this sentence.

line 301 it is questionable to suggest that large carabids favour complex vegetation because it provides cover from enemies. Some large species occur in open habitats with little vegetation. I would delete this suggestion, as it is rather speculative.

Authors: We have deleted this sentence.

lines 302-304 this sentence is also rather subjective and speculative. To justify the point that small features of such exposed habitats may provide shelter you could cite a study by Michiel Wallisdevries on butterflies and shelter provided by bushes. I believe it was his 2001paper in Restoration Ecology, though I am not certain. I am sure there are other suitable sources.

Authors: We have deleted this sentence.

line 309 in small ectothermic species, body temperature

Authors: We have added the coma.

line 316 in the studiedy ecosystems.

Authors: We have corrected the text.

line 324 habitat, temperatures are always

Authors: We have added the coma.

line 324 it is necessary to explain the point regarding predatory carabids here. Is there some connection with the effects of temperature?

Authors: The connection is possible following the study conducted by Frank and Bramböck 2016. Carabid feed more in high temperature, especially predatory species. We made this point more explicit.

line 328 companion study [16]

Authors: We have corrected the text.

showed that the carabid rarefied

Authors: We  have corrected the text.

 line 329 Even if the response for habitat type is similar to that for temperature, it needs to be explained clearly.

Authors: We have explained this point more clearly now.

lines 329-331 The information in this sentence also needs to be explained more clearly.

Authors: We have added a more explicit sentence.

lines 337-339 This should probably be incorporated into one of the research questions.

Authors: We have included this point in the Introduction when discussing the role of Vegetation characteristcs (point #5).

line 339 environment (i.e. the desert steppe), precipitation decreases

Authors: We have added the coma.

line 341 conditions (i.e. the typical steppe), the effect

Authors: We have added the coma.

line 342 If the authors feel that their results may be ‘in line for the so-called intermediate disturbance hypothesis’, then a vague suggestion like this is inadequate. They should include discussion about the hypothesis, preferably beginning with an introduction to the hypothesis in the Introduction section, and then careful consideration in the Discussion. A vague implication that the findings may be in keeping with the theory is not very useful in a scientific paper.

Authors: We have deleted this part.

lines 339-345 This sentence is far too long and complicated. I would recommend to break it up where there are now semi-colons, into three sentences.

Authors: Thank you, this part is now divided into three sentences.

line 347 Please check this. My understanding is that HB is appropriate when working with samples of a larger population and H’ when working with data of the whole population.

Authors: We have deleted the Shannon index resulsts, and hence we deleted this sentence.

line 348-351 Generally when working with pitfall data, one assumes that the data produced is an incomplete sample, though the justification provided here by the authors seems plausible.

Authors: Thank you. However, this sentence was no longer necessary, since we have deleted H’ results.

line 354 to climate factors, is more driven

Authors: We have added the comma.

line 355 I don’t understand what the authors mean by the term ‘homogenous negative effect of temperature’. In what way is the temperature effect homogenous?

Authors: We have deleted the term “homogenous” as it is not important to understand this part of the discussion.

We would like to thank once again the reviewer for her/his very detailed and useful comments, which helped us to ameliorate the manuscript. We hope that you are satisfied with this revision and that the manuscript is now in a suitable form for publication.

All the best,

Noelline Tsafack

(on behalf of all coauthors)
